# Influence of Ion Exchange Process Parameters on Broadband Differential Interference

**DOI:** 10.3390/s23136092

**Published:** 2023-07-02

**Authors:** Kazimierz Gut, Marek Błahut

**Affiliations:** Department of Optoelectronics, Silesian University of Technology, 2 Krzywoustego Str., 44-100 Gliwice, Poland; marek.blahut@polsl.pl

**Keywords:** waveguide interferometer, single-mode optical waveguide, ion exchange, gradient refractive profile

## Abstract

The paper presents theoretical analyses and experimental investigations of broadband differential interference in planar gradient waveguides made via K^+^-Na^+^ ion exchange in BK-7 glass. This technology, due to its large polarimetric dispersion, is especially useful for applications in differential interferometry. We discuss the influence of technological parameters on the operation characteristics of the structure in terms of sensor applications. The refractive index variation in the measured external surroundings affects the modal properties of TE and TM modes and the spectral distribution at the output of the differential interferometer. The optical system described in this work has been designed specifically for use in biological systems where variations in the index of refraction need to be measured.

## 1. Introduction

Interferometric planar waveguide sensors have great potential applications. Recent years have shown what a significant impact a viral epidemic can have on all human activity. Detecting viruses quickly is very important. The paper [1] shows the use of the Mach–Zehnder planar waveguide interferometer for influenza virus detection. As health is becoming more and more important to us, we want to detect hazardous substances. In [2], a polarization interferometer detecting zearalenone mycotoxins (ZONs) was described. Another interesting example is the measurement of the real and imaginary parts of the refractive index of a liquid in one single system. These tests were carried out using Young’s interferometric waveguide system [3]. In this study, the shifting of interference fringes and changes in their amplitudes were detected. In a typical guided wave interferometer, the light of one wavelength is usually propagated [4,5,6,7,8]. In [9], the use of a broadband Mach–Zehnder interferometer for monitoring changes in the refractive index was proposed. In this case, the interferometer light propagates from a certain wavelength range. The phase change in the measurement arm changes the spectral distribution at the output of the waveguide structure. The implemented interferometers of this type were described in [10,11,12,13,14,15]. The possibility of integrating a broadband light source, a planar interferometer and a spectral spectrum detection system on one substrate was also shown [16]. Recently, a paper [17] was published in which a broadband Mach–Zehnder interferometer enabled the detection of SARS-CoV-2 antibodies.

Broadband detection in a planar waveguide Young interferometer has also been proposed [18,19].

In [5], a division of waveguide interferometric systems into common- and double-path (single-channel and two-channel) systems was proposed. In a typical two-path interferometer, light propagates in the measurement path, where the measuring medium affects the phase of the beam, and in the reference path, which is isolated from this influence. The information about the phase change influenced by the measuring factor can be obtained from the recorded interference signal. In a co-path interferometer, the waveguide modes propagate along one path and the measurement factor affects the phase of all guided modes [20]. The modes may differ in polarization state—transversal electric (TE) and transversal magnetic transverse (TM) [21,22]—or mode number, for example, transversal electric zero (TE_0_) and transversal electric one (TE_1_) [23,24,25]. In this type of system, it is important that there is the greatest possible difference in sensitivity between the modes selected for interference. The signal recorded by the detector is a function of the cosine of the phase difference between the modes (since it is a periodic function, two different phases may occur at which the output light intensity is the same).

The paper in [26] describes spectropolarimetric interference in a planar waveguide.

The TE and TM modes in the visible wavelength range were introduced in the waveguide, and the transmitted spectrum was recorded at the output of the system with a spectrometer. A monotonic phase change (between modes) caused a monotonic shift of the recorded spectral distribution. The paper in [27] presents a model of a broadband interferometer for optical fibers with a Si_3_N^4^ waveguide layer on a SiO_2_ substrate.

The paper in [28] describes the phenomenon of differential interference and presents a model of a planar broadband differential interferometer based on the SU-8 polymer. On the basis of spectral measurements of the refractive indices of the waveguide and the substrate, the spectrometric signal of the system was determined for various refractive indices of the waveguide covering. The influence of geometrical parameters of the waveguide on the operation of the system is presented in [29]. The analysis of a broadband differential interferometer in a structure with a gradient refractive index profile is presented below.

## 2. Theoretical Analysis

The K^+^-Na^+^ ion-exchange technique in glass is a widely used method of producing passive integrated optical components [30,31,32,33,34,35]. Waveguide structures characterize low material loss and high thermal stability, and a small refractive index change distinguishes this process in single-mode applications. Modal properties of the gradient-index waveguides which decide the working characteristics of the difference interferometer can easily be controlled by the technological process conditions—the time of diffusion and the time of heating of the waveguides fabricated in the initial diffusion process—and by light propagation conditions.

The planar polarimetric interferometer analyzed in this work is based on a gradient-index waveguide made by K^+^-Na^+^ ion exchange in BK-7 glass surrounded by air.

The gradient index profile is the solution of the non-linear diffusion equation in BK-7 glass [33]:(1)$$\frac{\partial C}{\partial t}=\nabla \left[\frac{D}{1-\left(1-m\right)\cdot C}\nabla C\right]$$
where *C* is the concentration of introduced K^+^ ions, proportional to the refractive index changes ∆n; m is the mobility ratio of exchanged ions; and *D* is the diffusion coefficient.

It can be stated, on the basis of our experimental investigations, that ∆n = 0.0095 for TE polarization and 0.011 for TM polarization near the surface of BK-7 glass. The observed differences in the distribution of the refractive index for both orthogonal polarizations result from the anisotropy of strains taking place during the technological process, and these are of significant importance for the performance characteristics of the difference interferometer. The determined self-diffusion coefficient is equal to 2.18 m^2^/h at 400 °C for both profiles and *m* = 0.9 [36]. Refractive index distributions calculated numerically for TE polarization for the wavelength λ = 0.65 μm and the different times of diffusion denoted by the symbol t_D_ are shown in Figure 1.

In Figure 2, gradient index profiles for the both orthogonal polarizations TE and TM obtained for the time of diffusion t_D_ = 0.5 h are presented.

The analyzed structure is excited by a broadband light source from the range of 500 nm to 700 nm. Interference phenomena, observed in the examined gradient-index waveguide, result from its modal properties in the assumed spectral range of the broadband source. The modal properties of the numerically modeled waveguide structures were determined using an effective index method [37]. The dispersion characteristics of BK-7 glass were taken from [38]. During the diffusion process, the refractive index profile changes (Figure 1). The duration of the ion exchange process is the basic parameter that changes the conditions of light propagation in the waveguide structure. Figure 3 shows the effective refractive indexes of the modes for both TE and TM polarizations as functions of the process duration. The effective refractive indices for the 500 nm wavelength (the shortest wavelength in the considered range) are marked in red, and the 700 nm wavelength (the longest wavelength) in blue. The horizontal dashed lines indicate the refractive indices of the BK-7 substrate, which are different for different wavelengths due to material dispersion.

For a fixed wavelength, the value of the effective refractive index increases as the diffusion time increases. For applications in the broadband differential interferometer technology, the structure should be a single mode for both polarizations in the analyzed wavelength range. As the calculations show, this is achieved for the diffusion process time shorter than 0.8 h. Single-mode structures with diffusion times of 0.3 h, 0.4 h, 0.5 h, 0.6 h and 0.7 h were selected for further analysis. The values of the effective refractive indices for the selected diffusion times are the highest for a wavelength of 500 nm and decrease with increasing wavelength.

In Figure 3a,b, the ranges of changes in effective refractive indices for selected diffusion times are marked with arrows. To describe the phase relationships, the propagation constant *β* associated with the *N_eff_* relation is usually used:(2)$$\beta (\lambda )=\frac{2\pi Neff(\lambda )}{\lambda }$$

*β* expresses the phase change *ϕ* of the guided mode per unit path in the waveguide. For the propagation path length *l*, we can write:(3)ϕ(λ)=β(λ)·l

In differential interference, the phase difference between the orthogonal fundamental modes *TE* and *TM* is important, it can be expressed by the following relation:(4)Δϕ(λ)=Δβ(λ)·l
where
(5)$$\Delta \beta \left(\lambda \right)={\beta \left(\lambda \right)}_{TM}-{\beta \left(\lambda \right)}_{TE}$$

The determined differences in the propagation constants *Δβ*(*λ*) as a function of the wavelength for single-mode planar waveguides with different times of the ion exchange process are shown in Figure 4.

In the considered wavelength range, the difference Δβ(λ) decreases with increasing wavelength. The characteristics obtained are non-linear decreasing functions of the wavelength. Increasing the time of the ion exchange process almost does not change the shape of the characteristic but only causes its vertical shift to move towards higher values. Increasing the process time increases the value of Δβ for each wavelength in the considered range.

## 3. Experimental Verification

In order to verify the obtained dependence of the difference in the propagation constants on the wavelength, the measurement stand shown in Figure 5 was set up. The planar waveguide was made using the K^+^-Na^+^ ion exchange technique in BK7 glass. The process temperature was 400 °C and the time was 0.5 h. The glass substrate in which the planar waveguide was made had dimensions of 8 mm × 50 mm and a thickness of 2 mm.

The white light emitted by the LED diode (MWWHF2, Correlated Color Temperature 400K, Thorlabs Inc., Newton, NJ, USA)was fed into a multimode optical fiber (M44L022, Thorlabs Inc., Newton, NJ, USA) terminated with a collimator (F240 SMA-A, Thorlabs Inc., Newton, Thorlabs Inc., Newton, NJ, USA). Then, the light, after passing through the polarizer (LPVISE2X2 Thorlabs Inc., Thorlabs Inc., Newton, NJ, USA), fell on the input prism (SF14 glass) and was introduced into the planar waveguide obtained via the K^+^-Na^+^ ion exchange technique. The propagating light, after traveling a distance *l* in the waveguide (in this case, 28.7 mm), was output through the output prism (SF14 glass). The output beam, after passing through the polarizer (LPVISE2X2 Thorlabs Inc., Newton, Thorlabs Inc., Newton, NJ, USA), fell on the collimator (F240 SMA-A, THORLABS) connected to the spectrometer (HR4000 CG–UV-VIS-NIR, Ocean Optics) with a multimode optical fiber (M28L02, Thorlabs Inc., Newton Thorlabs Inc., Newton, NJ, USA). In this way, the optical spectrum at the output of the system could be recorded.

The polarizer at the input of the system enabled (by rotating around the axis) an even division of the optical power between the orthogonal basic modes TE_0_ and TM_0_. The rotation of the output polarizer enabled the recording of the following mode spectra: TE_0_ (transmission axis of the polarizer is parallel to the surface of the waveguide—0°), TM_0_ (transmission axis of the polarizer is rotated by 90°) or the spectra of both modes simultaneously (transmission axis of the polarizer is rotated by 45°). These three recorded spectra are shown in Figure 6.

During the registration of the spectrum, when light from the TE_0_ and TM_0_ modes reached the spectrometer, oscillations in the registered signal were observed. Orthogonal modes have different phase velocities for a given wavelength. After bringing the light from the TE_0_ and TM_0_ modes to one polarization plane (the transmission axis of the polarizer was rotated by 45°), an interference signal was recorded. The interference signal itself after subtracting the arithmetic mean of the TE_0_ and TM_0_ modes’ spectra is shown in Figure 7. The length of the propagation path in the waveguide was 27.8 mm.

In Figure 7, the successive interference extremes are marked with arrows. The extreme for which the difference in propagation constant Δβ is the smallest was adopted as the first one. Let us denote the wavelength corresponding to the first extremum of the extreme as λ_1_; similarly, we will assume the wavelengths corresponding to the subsequent extremes in Figure 7 are λ_2_, λ_3_, λ_4_ ……, etc. As shown earlier, the difference in propagation constants Δβ decreases with increasing wavelength. Therefore, if λ_1_ > λ_2_ > λ_3_ > λ_4_ ……, then Δβ(λ_1_) < Δβ(λ_2_) < Δβ(λ_3_) < Δβ(λ_4_) ……, etc.

For the first extreme (in this case, the maximum) occurring at wavelength λ_1_, the phase difference between the TE and TM modes *Δϕ* is equal to an integer multiple of 2π. For the second extreme (in this case, the minimum) occurring at the wavelength λ^2^, it increases by π. So, we can write the following relations for the successive extremes:*Δϕ*(*λ*_1_) = 2*π C**Δϕ*(*λ*_2_) = *Δϕ*(*λ*_1_) + *π* = 2*π C*+ *π**Δϕ*(*λ*_3_) = *Δϕ*(*λ*_2_) + *π* = 2*π C* + 2*π**Δϕ*(*λ*_4_) = *Δϕ*(*λ*_3_) + *π* = 2*π C* + 3*π*……………………(6)

Using relationship (4), (denoting *l* as the length of the propagation path in the waveguide), we obtain the following relations:(7)$$\Delta \beta \left(\lambda_{1}\right)\times l=2\pi C\;\Delta \beta \left(\lambda_{2}\right)\times l=\left[\Delta \beta \left(\lambda_{1}\right)\times l\right]+\pi \;\Delta \beta \left(\lambda_{3}\right)\times l=\left[\Delta \beta \left(\lambda_{1}\right)\times l\right]+2\pi \;\Delta \beta \left(\lambda_{4}\right)\times l=\left[\Delta \beta \left(\lambda_{1}\right)\times l\right]+3\pi \;\ldots \ldots \ldots \ldots \ldots \ldots \ldots \ldots$$

After dividing both sides by *l*, we obtain the values of the difference of the propagation constants for the wavelengths corresponding to the extremes:(8)$$\Delta \beta \left(\lambda_{1}\right)=2\pi C/l\;\Delta \beta \left(\lambda_{2}\right)=\Delta \beta \left(\lambda_{1}\right)+\pi /l\;\Delta \beta \left(\lambda_{3}\right)=\Delta \beta \left(\lambda_{1}\right)+2\pi /l\;\Delta \beta \left(\lambda_{4}\right)=\Delta \beta \left(\lambda_{1}\right)+3\pi /l\;\ldots \ldots \ldots \ldots \ldots \ldots \ldots \ldots$$

In this way, the interference spectrum of orthogonal modes allows for the determination of the *Δβ*(*λ*) function with an accuracy of an unknown constant. By adjusting this constant on the basis of previously determined theoretical characteristics, it is possible to compare experimental results and theory. Figure 8 shows the measurement points applied to the previously obtained theoretical dependence.

Very good agreement between the experimental data and the performed modeling was obtained.

## 4. Broadband Interferometer as Sensors

The graphs and measurements presented above concerned the waveguide structure made using the ion exchange technique in glass when the waveguide was surrounded by air. The obtained characteristics have very good agreement with the measurement data. Therefore, it was decided to consider the operation of the described structure as a broadband differential interferometer monitoring changes in the refractive index of the optical path cover. It was assumed that the optical system will be used for the analysis of biological substances. For this reason, refractive index changes in the cover refer to the water solutions and can be expressed as $n_{H_{2}O}+∆n_{c}$, with $∆n_{c}$= (0 ÷ 0.015). Verified experimental refractive profiles for both polarizations were used to determine the relationships presented below. We only changed the value of the refractive index of the refraction waveguide covering, taking its dispersion into account.

If light with a known spectral distribution *I_in_*(*λ*) and the same intensity for both orthogonal modes is introduced into the waveguide, the spectral distribution *I_out_*(*λ*) will be obtained at the output. The modification of the refractive index of the cover *n_c_* changes the propagation conditions of both modes in a different way, which leads to a change in the phase difference *Δϕ* between them. At the output, the spectral signal is a function of the refractive index of the cover *n_c_*, the wavelength and the length of the propagation path in the waveguide *l*:*I_out_*(*λ*) = *Τ*(*n_c_*, *λ*, *l*) × *I_in_* (*λ*), (9)
where *T*(*n_c_*, *λ*, *l*) means
*Τ* (*n_c_*, *λ*, *l*) =1/2 × {1 + *cos*[*Δβ*(*n_c_*, *λ*) × l]}(10)

Figure 9 shows the functions *T*(*n_c_*, *λ*, *l* = 0.5 cm) for the propagation path length of 0.5 cm. The red color shows the value of *T*(*n_c_*, *λ*, *l*) if the waveguide coating is water, and the blue color corresponds to an increase in the value of *n_c_* by 0.015. As the refractive index increases, the characteristic shifts towards a longer wavelength (red shift occurs).

Figure 10 presents the function T(n_c_, λ, *l* = 1.0 cm) for a propagation length of 1 cm. As in the previous figure, the red color shows the dependence for the refractive index of water, and the blue color shows the dependence when the refractive index of the coating increases by 0.015. The number of extremes depends on the length of the propagation path in the waveguide. Increasing the length of the propagation path causes an increase in the number of extremes. In this way, the output signal can be modified.

The characteristic shown in Figure 11 is an extension (magnifying glass) of the characteristic of the propagation path for *l* = 1 cm in the wavelength range of 0.65 um to 0.7 um. The black color represents the value of T(n_c,_ λ) if the waveguide is covered with water; the other colors correspond to the increase in the refractive index of the coverage by 0.005, 0.010 and 0.015, respectively. The increase in the refractive index value corresponds to the shift in the extremes of the function T(n_c_, λ, *l*). The wide availability of fiber optic spectrometers allows the use of the proposed method of changing the refractive index of cover.

## 5. Conclusions

The paper presents the characteristics of a planar broadband differential interferometer with a gradient refractive index profile. The numerical data obtained were experimentally verified. It was also shown how the output signal changes when the refractive index changes in the aqueous environment, which is of particular importance in the design of chemical and biochemical sensors. The change in the output characteristics of the sensor applies to the entire spectral range, as opposed to resonant-type sensors, where a change is only observed in a certain range of the spectrum. In the case of ion exchange, the time of the process practically does not change the course of the Δβ(λ) characteristics (Figure 4); it only increases their value. As shown in [29], the derivative of the Δβ(λ) function depends on the number of extremes and the change in their location when the refractive index changes.

## Figures and Tables

**Figure 1 sensors-23-06092-f001:**
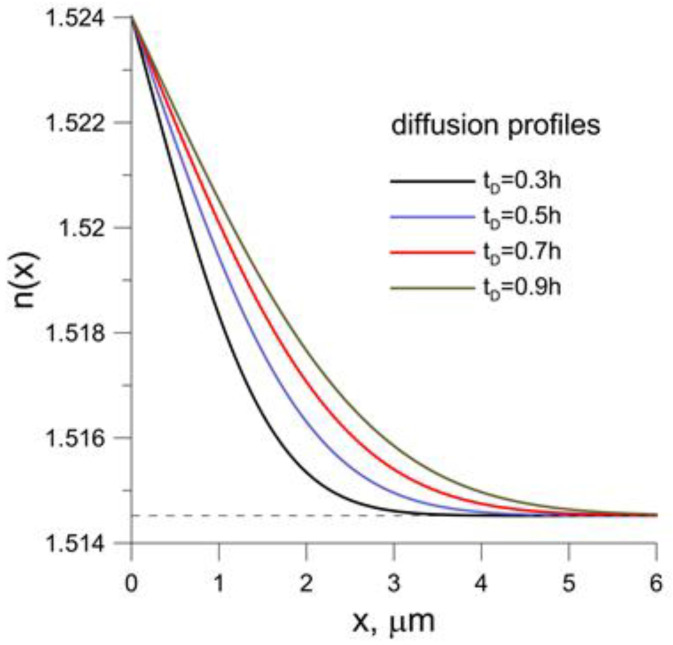
Refractive index distribution $n\left(x\right)$ of gradient index waveguides for TE polarization, for the different times of diffusion. The wavelength λ=650 nm. Dashed line denotes the refactive index of BK-7 substrate.

**Figure 2 sensors-23-06092-f002:**
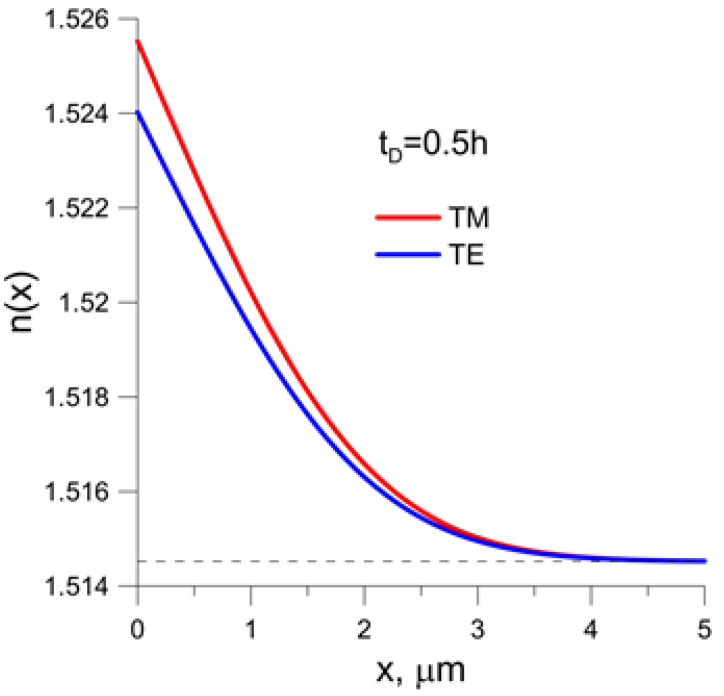
Gradient index profiles for the both orthogonal polarizations TE and TM obtained for the time of diffusion t_D_ = 0.5 h. Dashed line denotes the refactive index of BK-7 substrate.

**Figure 3 sensors-23-06092-f003:**
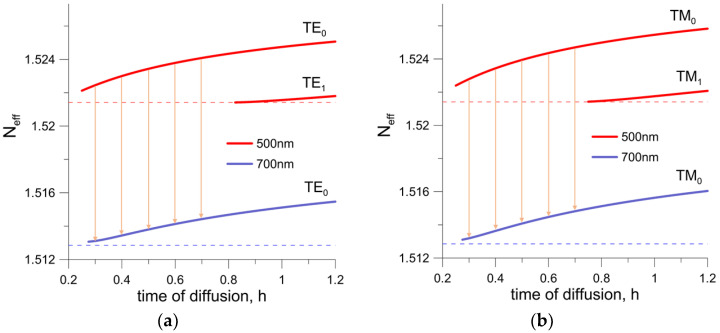
Numerically calculated dependences of effective refractive indices of TE modes (**a**) and TM modes (**b**) on the time of diffusion process for the wavelengths of 500 nm and 700 nm. Dashed line denotes the refactive index of BK-7 substrate.

**Figure 4 sensors-23-06092-f004:**
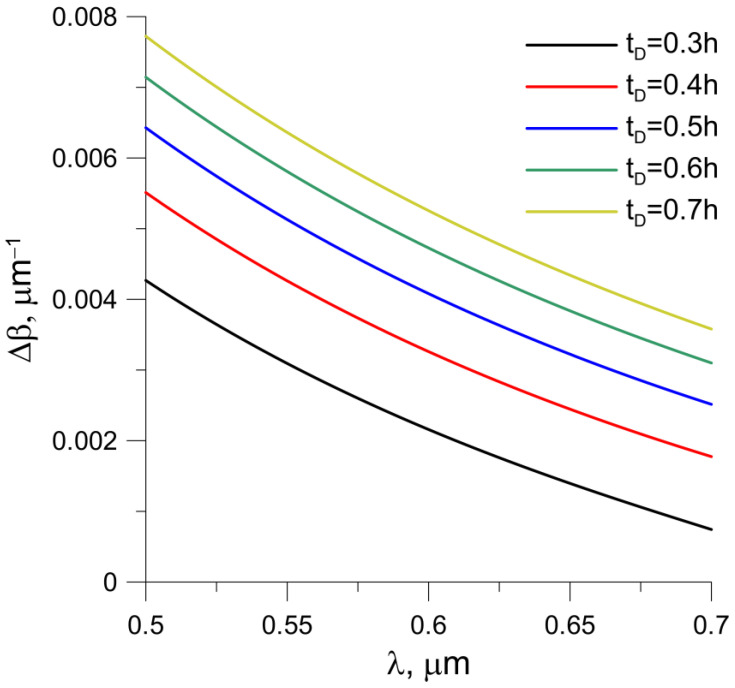
The difference in propagation constants of TM and TE modes as a function of wavelength for different times of diffusion.

**Figure 5 sensors-23-06092-f005:**
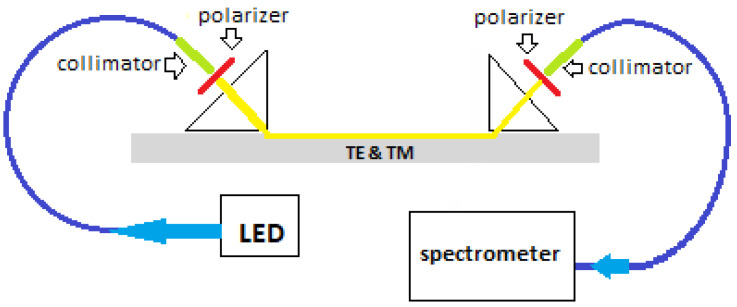
Test stand for introducing white light into a planar waveguide and measuring the spectral distribution at the output of the interferometer.

**Figure 6 sensors-23-06092-f006:**
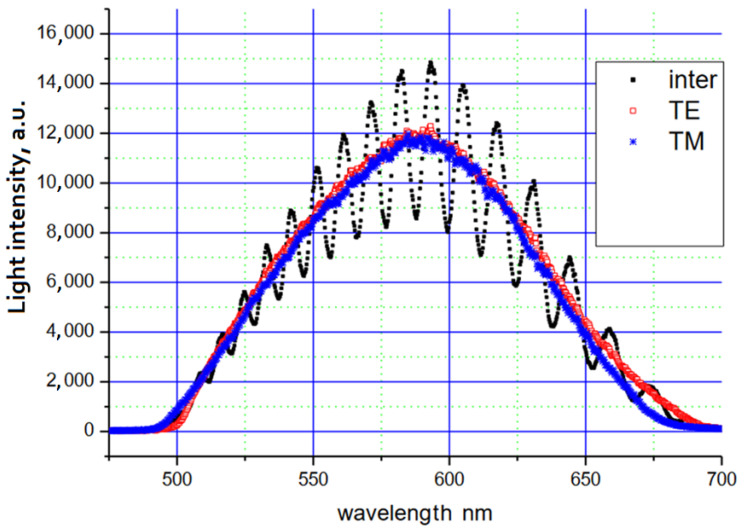
Spectra recorded at different angular settings of the transmission axis of the output polarizer.

**Figure 7 sensors-23-06092-f007:**
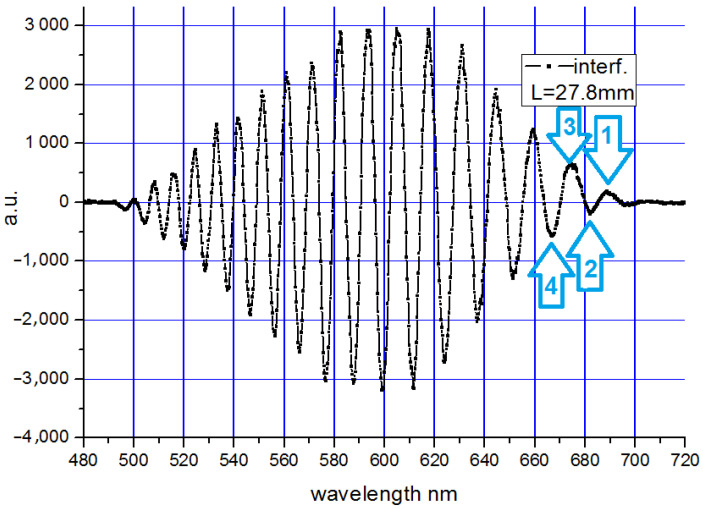
Interference signal for propagation length (distance between prisms) of 28.7 mm.

**Figure 8 sensors-23-06092-f008:**
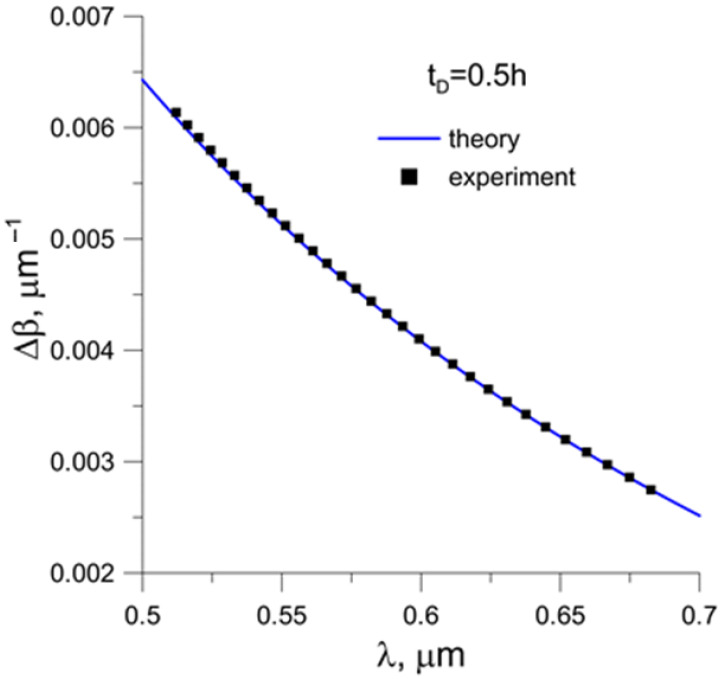
Comparison of numerically determined data with the experiment.

**Figure 9 sensors-23-06092-f009:**
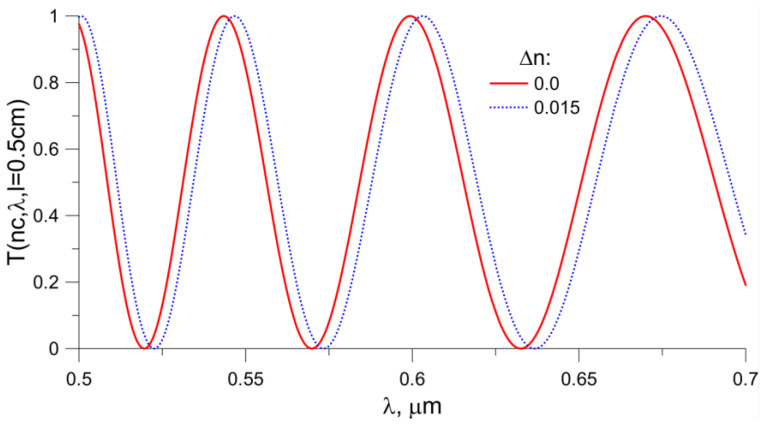
*T*(*n_c_*_,_ *λ*, *l* = 0.5 cm) function. The red color corresponds to the situation when water is covering the waveguide, and the blue color corresponds to an increase in the *n_c_* value by 0.015. The length of the propagation path *l* was assumed to be 0.5 cm.

**Figure 10 sensors-23-06092-f010:**
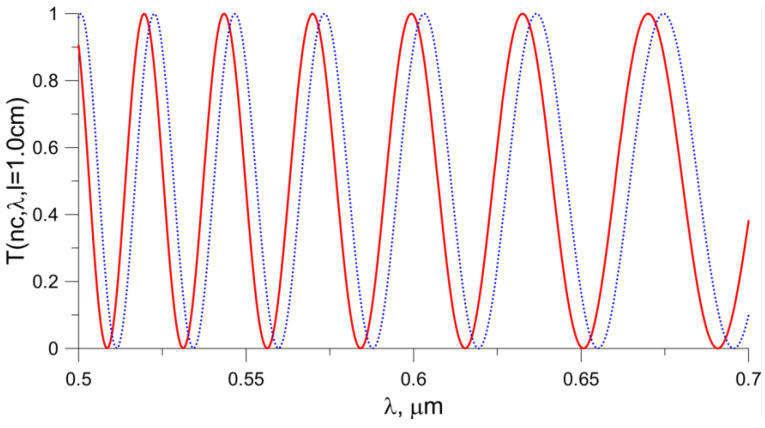
*T*(*n_c_*, *λ*, *l* = 1.0 cm) function. The red color corresponds to the situation when water is covering the waveguide, and the blue color corresponds to an increase in the n_c_ value by 0.015. The length of the propagation path *l* was assumed to be 1.0 cm.

**Figure 11 sensors-23-06092-f011:**
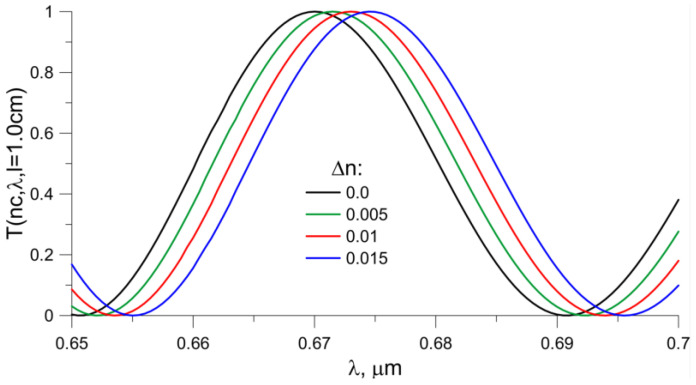
Function *T*(*n_c_*, *λ*, *l* = 1.0 cm). The black color corresponds to the situation when water is the coating of the waveguide, and the other colors correspond to the higher value of the coating fracture index n_c_ by 0.005, 0.010 and 0.015, respectively. The length of the propagation path l is 1.0 cm.

## Data Availability

Not applicable.

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
