# Peer review of "Influence of Ion Exchange Process Parameters on Broadband Differential Interference"

_sensors, 2023, doi:10.3390/s23136092_

Round 1

Reviewer 1 Report

The paper deals with the theoretical and experimental investigation of the influence of the gradient refractive index profile achieved by controlled ion exchange on the propagation constant dispersion. The structure of the paper is logical and the main idea is easy to follow. The results obtained are sound and convincing. However, the text of the manuscript contains too many typos that must be removed:

See:

Page 1, line 33 (Z In the work...)

Page 2, line 63 (...stability. and...)

Page 4, line 125 and 127 - Please, unify the font for the symbol "l" (here and throughout the text, e.g. line 204, 249, 266) which stands for the path length

Page 6, lines 162-164, and 171 - Please, unify the writing of TE0, TM0 and TM

Page 7, line 189 - Number 2 next to the "lambda" symbol should be the subscript, not the superscript.

Page 10, line 284 - Please, remove the extra space at the end of the sentence (...changes  .)

In addition, I have some comments and questions.

1. Page 2, lines 76 and 77 - It is not quite clear whether the statement is based on the results obtained within this work or in any other work. Please, specify.

2. Page 3, Fig.1 and Fig.2, and Page 4., lines 108 and 109 - One of the parameters determining the single mode regime of the waveguide for the given wavelength is the waveguide thickness. The authors do not mention explicitly this parameter. Do they consider the diffusion depth (x coordinate) as the (effective) waveguide thickness?

3. Page 7, line 197 - I think, here is more appropriate to refer to Eq.(4) than to Eq. (3).

4. Page 9

lines 242-244 - Please, rethink of the sentence

"Black colour shows..." - There is no black coloured curve in the Figure 9.

"... blue colour increases the value of nc..." - How can a blue colour increase the value of refractive index?

According to the text of the manuscript, Figures 9 and 10 show the function T(nc, lambda), but the axis description says it is "Output power distribution" in arbitrary units. What is true?

Despite the overall good level of English, I found some language issues:

Page 4:

lines 101, and 102 - I recommend using "indices" instead of "indexes"

line 107 - Please, check the grammar of this sentence (singular vs plural)

Page 9:

line 251 - Please, check the grammar: "... water is cover the waveguide..."

Author Response

Thank you very much for your thorough review.

The reviewer writes:

“However, the text of the manuscript contains too many typos that must be removed:

Page 1, line 33 (Z In the work...)

Page 2, line 63 (...stability. and...)

Page 4, line 125 and 127 - Please, unify the font for the symbol "l" (here and throughout the text, e.g. line 204, 249, 266) which stands for the path length

Page 6, lines 162-164, and 171 - Please, unify the writing of TE0, TM0 and TM

Page 7, line 189 - Number 2 next to the "lambda" symbol should be the subscript, not the superscript.

Page 10, line 284 - Please, remove the extra space at the end of the sentence (...changes  .)”

Removed typos according to reviewer's comments

The reviewer writes:

“In addition, I have some comments and questions.

  1. Page 2, lines 76 and 77 - It is not quite clear whether the statement is based on the results obtained within this work or in any other work. Please, specify.”

We changed the above sentence to:

It can be stated on the base of our experimental  investigations, that ∆n=0.0095 for TE polarization and 0.011 for TM polarization near the surface of BK-7 glass.

The reviewer writes:

“2. Page 3, Fig.1 and Fig.2, and Page 4., lines 108 and 109 - One of the parameters determining the single mode regime of the waveguide for the given wavelength is the waveguide thickness. The authors do not mention explicitly this parameter. Do they consider the diffusion depth (x coordinate) as the (effective) waveguide thickness?”

It is difficult to determine the waveguide thickness in the case of gradient index waveguides.  For diffusion waveguides it may be the diffusion depth, defined as sqtr(D*t), but the single-mode regime of gradient-index waveguides depends also on the shape of refractive index distribution (concave or convex).  The best way is to connect the modal properties of gradient waveguides with gradient index profile n(x) with  coordinate x in um perpendicular to the waveguide surface.  

The reviewer writes:

“3. Page 7, line 197 - I think, here is more appropriate to refer to Eq.(4) than to Eq. (3).”

Pattern reference number changed.

The reviewer writes:

”4. Page 9

lines 242-244 - Please, rethink of the sentence

"Black colour shows..." - There is no black coloured curve in the Figure 9.

"... blue colour increases the value of nc..." - How can a blue colour increase the value of refractive index?”

Appropriate colors have been introduced into the text and the sentence has been corrected:

Red colour shows the value of T(nc, λ, l) if the waveguide coating is water, blue colour corresponds to an increase the value of nc by 0.015.

The reviewer writes:

“According to the text of the manuscript, Figures 9 and 10 show the function T(nc, lambda), but the axis description says it is "Output power distribution" in arbitrary units. What is true?”

The description of the vertical axis in the mentioned drawings has been changed. Vertical axis strings are now T(nc, lambda, l

The reviewer writes:

“Comments on the Quality of English Language

Despite the overall good level of English, I found some language issues:

Page 4:

lines 101, and 102 - I recommend using "indices" instead of "indexes" ”

The word proposed by the reviewer has been used.

The reviewer writes:

“line 107 - Please, check the grammar of this sentence (singular vs plural)”

The sentence after the correction is as follows:

For a fixed wavelength, the value of the effective refractive  index increases as the diffusion time increases.

The reviewer writes:

“Page 9:

line 251 - Please, check the grammar: "... water is cover the waveguide..." “

The sentence after the correction is as follows:

…water is covering the waveguide…

Again, thank you for your thorough review.

 Authors

Reviewer 2 Report

In this manuscript, Gut et al. report the influence of ion exchange parameters on broadband differential interference. The following comments need to be addressed.

Comment 1. (Line 20) The authors should list some potential applications of Interferometric planar waveguide sensors and provide the related references.

Comment 2. (Line 33) Delete Z.

Comment 3. (Line 46) Fix the sentence.

Comment 4. Terms should be abbreviated when they first appear (TE, TM, etc.). TE0, TE1, TM0, TM1, tD …. should be clearly defined when they first appear in the manuscript.

Comment 5. Check all the subscripts and superscripts, upper case and lower case. Example: line 162, line 171 ….

Comment 6. (Section 4. Broadband interferometer as sensors) I think this part is a theoretical analysis and proposal. Did you apply the broadband interferometer as sensors experimentally? If so, those data should be included. Otherwise, the application is not convincing.

Comment 7. To ensure the reproducibility of the work, the experimental details should be provided, including detailed information of the equipment and instruments. For example, LED type, parameters, and brand, spectrometer type and brand, BK7 glass dimensions and its related information, etc. 

Moderate editing is needed.

Author Response

Thank you very much for your thorough review.

Below are the answers to the review in points.

Comment 1. (Line 20) The authors should list some potential applications of Interferometric planar waveguide sensors and provide the related references.

Our answer:

An excerpt has been added in the text:

Recent years have shown what a significant impact a viral epidemic can have on the entire human activity. Detecting the virus quickly is very important. The paper [1] shows the use of the Mach-Zhender planar waveguide interferometer for influenza virus detection. As health is becoming more and more important to us, we want to detect hazardous substances. In [2], a polarization interferometer detecting zearalenone mycotoxins (ZON) was described. Another interesting example is the measurement of the real and imaginary parts of the refractive index of a liquid in one single system. The tests were carried out using Young's interferometric waveguide system [3]. In this study, the shifting of interference fringes and changes in their amplitudes were detected.

……

  1. Sakamoto, H.; Minpou, Y.; Sawai,T.; Enami, Y.; Suye, S.; A Novel Optical Biosensing System Using Mach–Zehnder-Type Optical Waveguide for Influenza Virus Detection. Appl. Biochem. Biotechnol. 2016, 178, 687–694.
  2. Nabok, A.; Al-Jawdah , A.M.; Gémes B.; Takács, E.; Székács , A. An Optical Planar Waveguide-Based Immunosensors for Determination of Fusarium Mycotoxin Zearalenone. Toxins 2021, 13, 89.
  3. Zhou, C.; Hedayati,M. K.; Kristensen, A. Multifunctional waveguide interferometer sensor: simultaneous detection of refraction and absorption with size-exclusion function. Express 2018, 26, 24372-24383.

Comment 2. (Line 33) Delete Z.

Our answer:

The letter Z has been removed

Comment 3. (Line 46) Fix the sentence.

Our answer:

Sentence changed (Line 46):

Paper [26] describes spectropolarimetric interference in a planar waveguide.

Comment 4. Terms should be abbreviated when they first appear (TE, TM, etc.). TE0, TE1, TM0, TM1, tD …. should be clearly defined when they first appear in the manuscript.

Our answer:

Sentences changed:

The modes may differ in polarization state: transversal  electric  (TE) and transversal magnetic  (TM) [21, 22] or mode number for example; transversal electric zero (TE0) and transversal electric one (TE1) [23-24].

….

Refractive index distributions calculated numerically for TE polarization  for the wavelength l=0.65mm and the different times of diffusion denoted by the symbol tD are shown in the Fig.1.

Comment 5. Check all the subscripts and superscripts, upper case and lower case. Example: line 162, line 171 ….

Our answer:

Corrected incorrect superscripts and subscripts

Comment 6. (Section 4. Broadband interferometer as sensors) I think this part is a theoretical analysis and proposal. Did you apply the broadband interferometer as sensors experimentally? If so, those data should be included. Otherwise, the application is not convincing.

Our answer:

In our work, we want to show the further perspective and possibilities of the described structure. Therefore, it was decided to present the operation of the system in an aqueous environment typical for biological applications. It was described how the output signal will change with changes in the refractive index in this environment. It seems to us that the presented considerations will be an inspiration to undertake further strictly experimental work.

 To avoid confusion in the work, the following sentences have been added:

“Verified experimental refractive profiles for both polarizations were used to determine the relationships presented below. Changed only the value of the refractive index of the refraction waveguide covering, taking into account its dispersion.”

Comment 7. To ensure the reproducibility of the work, the experimental details should be provided, including detailed information of the equipment and instruments. For example, LED type, parameters, and brand, spectrometer type and brand, BK7 glass dimensions and its related information, etc.

Our answer:

Added catalog specifications and manufacturer in the description of the measuring station.

“The glass substrate in which the planar waveguide was made had dimensions of 8mm x 50mm and a thickness of 2mm

The white light emitted by the LED diode (MWWHF2 Correlated Color Temperature 400K, THORLABS) was fed into a multimode optical fiber (M44L022, THORLABS) terminated with a collimator (F240 SMA-A, THORLABS). Then, the light, after passing through the polarizer (LPVISE2X2 THORLABS), falls on the input prism (SF14 glass) and is introduced into the planar waveguide obtained by the K+-Na+ ion exchange technique. The propagating light after traveling a distance l in the waveguide (in this case 28.7 mm) is output through the output prism (SF14 glass). The output beam, after passing through the polarizer (LPVISE2X2 THORLABS)), falls on the collimator (F240 SMA-A, THORLABS) connected to the spectrometer (HR4000 CG–UV-VIS-NIR, Ocean Optics) with a multimode optical fiber (M28L02, THORLABS). In this way, the optical spectrum at the output of the system can be recorded. “

Manuscript style and editing errors have been corrected.

Thank you for your thorough and informative review.

 Authors

Reviewer 3 Report

This study focuses on the operational characteristics of a planar, broadband, differential optical interferometer. The sensing element is a graded index waveguide which is modified by ion-exchange brought about by diffusion. This exchange modifies the operational characteristics of the interferometer and changes its ability to sense variations in the environment which might be biochemical in nature. The paper is well organized, contains a range of interesting results (computed and measured) and should be of interest to researchers developing interferometric sensors for liquid environments.

In Figure 6, the graph label "wavelength" is misspelled and the vertical axis needs to be labelled.

In Figure 7, the graph label "wavelength" is misspelled.

Even though the English-usage doesn't obscure the meaning of various sentences, the structure of many sentences is awkward and could be improved. For example, in the Abstract, the authors write, "The optical system described will be designed to the analysis of biological substances." Maybe this could be written, "The optical system described in this work has been designed specifically for use in biological systems where variations in index of refraction need to be measured."?

Author Response

Thank you very much for your thorough review.

Below are the answers to the review

The reviewer writes:

In Figure 6, the graph label "wavelength" is misspelled and the vertical axis needs to be labelled.

In Figure 7, the graph label "wavelength" is misspelled.

Our answer:

The description of the vertical axis in Fig. 6 has been added, the description under the horizontal axis has been corrected. In Figure 7, the description under the horizontal axis has been corrected.

Comments on the Quality of English Language

Even though the English-usage doesn't obscure the meaning of various sentences, the structure of many sentences is awkward and could be improved. For example, in the Abstract, the authors write, "The optical system described will be designed to the analysis of biological substances." Maybe this could be written, "The optical system described in this work has been designed specifically for use in biological systems where variations in index of refraction need to be measured."?

Our answer:

The abstract was changed as suggested by the reviewer.

Manuscript style and editing errors have been corrected.

Thank you for spotting our errors in the manuscript.

 Authors

Round 2

Reviewer 2 Report

I recommend this manuscript for publication.

Minor editing is needed.